# The Multifaceted Syndromic Primary Immunodeficiencies in Children

**DOI:** 10.3390/jcm12154964

**Published:** 2023-07-28

**Authors:** Khuen Foong Ng, Anu Goenka, Florence Manyika, Jolanta Bernatoniene

**Affiliations:** 1Department of Paediatric Immunology and Infectious Diseases, Bristol Royal Hospital for Children, Bristol BS1 3NU, UK; anu.goenka@bristol.ac.uk (A.G.); florence.manyika@uhbw.nhs.uk (F.M.); jolanta.bernatoniene@uhbw.nhs.uk (J.B.); 2Cellular and Molecular Medicine, University of Bristol, Bristol BS8 1TD, UK

**Keywords:** primary immunodeficiency disorder, genetic syndromes, syndromic children, multisystem disorder, inborn errors of immunity

## Abstract

Background: Disorders of immunity are poorly recognised in some rare multisystem genetic conditions. We aim to describe syndromic features and immunological defects in children with syndromic primary immunodeficiencies (sPIDs). Methods: This is a retrospective descriptive study of children aged 0–18 years with sPIDs under the care of the paediatric immunology service at the Bristol Royal Hospital for Children, United Kingdom, from January 2006 to September 2021. Results: sPIDs were identified in 36 patients. Genetic diagnoses which are not commonly associated with PIDs and not included in the International Union of Immunological Societies classification were present in 7/36 (19%): Trisomy 22, Arboleda-Tham syndrome, 2p16.3 deletion syndrome, supernumerary ring chromosome 20 syndrome, Myhre syndrome, Noonan syndrome, and trichothiodystrophy/Cockayne syndrome complex. Recurrent and/or severe infections were the most common clinical features (n = 33, 92%). Approximately half had combined immunodeficiency or antibody deficiency. The most common extra-immunological manifestations include dysmorphism (72%), disorders of nervous (78%), musculoskeletal (69%), haematology/lymphatic (58%), and gastrointestinal, hepatic/pancreatic (58%) systems. Conclusions: Patients with sPIDs often have multiorgan involvement and some are non-immunologically mediated. There should be a low threshold to clinically assess and investigate for disorders of immunity in any patients with syndromic features especially when they present with recurrent/severe/opportunistic infections, features of immune dysregulation, autoinflammation or lymphoproliferation.

## 1. Introduction

Syndromic primary immunodeficiencies (sPIDs) are conditions with features of inborn errors of immunity (IEI) overlapping with other multisystem clinical manifestations, which are not directly associated with the immunologic deficit [1]. Well-described examples include Wiskott–Aldrich syndrome (WAS), ataxia telangiectasia (AT) and DiGeorge syndrome (DGS). The molecular aetiology of these conditions ranges from single gene defects to chromosomal anomalies, and can help us to understand the pathobiology underlying the immunodeficiency, organ dysfunction and dysmorphism. The increasing availability and use of next generation sequencing (NGS) technology such as whole exome/genome sequencing (WES/WGS) has resulted in the rapid molecular characterisation of previously defined and undefined genetic disorders, including those of sPIDs [2]. For example, WGS revealed IVNS1ABP haploinsufficiency, MIM #618969 (confirmed by Western blot), in four patients who had long-standing multiple warts, cutaneous boils, recurrent nasal polyposis and sinusitis, bronchiectasis, inflammatory colitis, protracted Epstein-Barr virus (EBV) infection, CD4 and CD8 T cell as well as B cell lymphopaenia, hypogammaglobulinaemia, coeliac disease and retinal vasculitis, as well as non-immunological features such as achalasia and hypothyroidism [3].

Cohort descriptions of sPIDs are lacking in the literature and disorders of immunity are poorly recognised in some rare multisystem genetic conditions [4,5,6,7]. In contrast, primary immunodeficiencies (PIDs) in association with infections, inflammation, autoimmunity, allergy, lymphoproliferation and malignancy are widely reported. In order to raise awareness and better describe the association between syndromic features and immunological defects, here we describe children with a wide spectrum of sPIDs diagnosed and followed up in our institution over a period of 15 years, including genetic disorders, which have been rarely associated with PIDs.

## 2. Materials and Methods

We performed a retrospective case series description of all children aged 0–18 years with sPIDs under the care of the paediatric immunology service at the Bristol Royal Hospital for Children, United Kingdom, from January 2006 to September 2021. This department serves a population of over 5 million individuals residing in Southwest region of England, United Kingdom. Clinical data (demographics, clinical characteristics, treatments and outcomes) were identified from a local clinical database that includes all children with immunodeficiency. sPIDs were defined as conditions with features of immunodeficiencies overlapping with other multisystem clinical manifestations that are not always associated with an immunologic deficit [1]. Inclusion criteria are:
At least one of the following:
Dysmorphic features such as short stature, facial abnormalities, microcephaly, and skeletal abnormalities;Other organ manifestations such as albinism, hair or tooth abnormalities, heart or kidney defects, hearing abnormalities, primary neurodevelopmental delay, and seizures.
AND
2. At least one numeric or functional abnormal finding upon immunological investigation.
AND
3. Exclusion of secondary causes for immunological abnormalities (infection and malignancy).
Disorders of immunity were classified according to European Society for Immunodeficiencies PID definitions [1].


## 3. Results

During the study period, we identified 36 children meeting inclusion criteria of sPID: 22 males, 61%, and 14 females, 39%. The median age was 11.4 years (range 0.6–26.8 years). IEI (n = 36) and genetic (n = 33) diagnoses were made at a median age of 5 years (range 0.2–17 years) and 2 years (0–16 years), respectively.

A summary of the immunological disorders and their treatment alongside genetic diagnoses is displayed in Table 1. There were 17 different genetic syndromes associated with PIDs identified. Genetic diagnoses that are not commonly associated with PIDs and not included in the International Union of Immunological Societies classification were present in 7/36 children (19%): Trisomy 22, Arboleda-Tham syndrome, 2p16.3 deletion syndrome, supernumerary ring chromosome 20 syndrome, Myhre syndrome, Noonan syndrome, and trichothiodystrophy/Cockayne syndrome complex. 

Approximately half of the children had a combined immunodeficiency or antibody deficiency (Figure 1). Recurrent/severe infections were the most common clinical feature (33/36, 92%). The majority of patients presented with infections before being one year old (20/33; 61%), 8/33 (24%) between one and five years, and 5/33 (15%) after five years. Nine (25%) patients had bacterial sepsis. Bronchiectasis occurred in 14% of patients (5/36): DGS (n = 3); Kabuki syndrome (n = 1) and cartilage hair hypoplasia (CHH) (n = 1). The most common extra-immunological manifestations included dysmorphism (72%), disorders of nervous (78%), musculoskeletal (69%), haematology/lymphatic (58%), and gastrointestinal, hepatic/pancreatic (58%) systems. Genetic conditions associated with chronic diarrhoea (n = 8, 22%) were trichohepatoenteric syndrome (n = 2), sideroblastic anemia with B cell immunodeficiency, periodic fevers and developmental delay (SIFD) (n = 3), supernumerary ring chromosome 20 syndrome (n = 1), CHH (n = 1) and WAS (n = 1). Lymphoma developed in two AT patients (6%).

Antibiotic prophylaxis was started in 25 (69%) patients, immunoglobulin replacement in 30 (83%) patients, and biologic modifier therapy in 10 (28%) patients. Three children (8%) underwent hematopoietic stem cell transplantation (HSCT): WAS (n = 2) and SIFD (n = 1). One patient with WAS had autologous gene-corrected HSCT and one DGS patient received a thymic transplant. Full descriptions of clinical characteristics, treatments and outcomes are available in the online Appendix A. 

## 4. Discussion

We describe diverse immunological and non-immunological manifestations (Figure 1) of a cohort of children with sPID. One fifth of our patients had genetic disorders, which were very rarely associated with IEI but not included in the International Union of Immunological Societies (IUIS) classification: Trisomy 22 and Arboleda-Tham syndrome, 2p16.3 deletion syndrome, supernumerary ring chromosome 20 syndrome, Myhre syndrome, Noonan syndrome, and trichothiodystrophy/Cockayne syndrome complex [2]. Our findings raise the possibility that there could be children attending specialty clinics with rare genetic conditions who have unrecognized disorders of immunity, which is of potential clinical significance to a wide range of healthcare professionals including paediatricians, subspecialists and geneticists. Patients with IEI do not only have a tendency to develop infections but also autoinflammation, autoimmunity, lymphoproliferation, malignancy and allergy [8]. We advocate that such children undergo appropriate immunological investigations if clinically indicated. Immunologists should also look beyond the 485 genetic defects described in the 2022 IUIS document as some of these rare genetic disorders with immunodeficiencies were not included in the classification [2]. 

Our cohort of patients with sPIDs exhibited a variety of immunological defects predisposing them to recurrent infections, autoinflammation, autoimmunity, immune dysregulation, lymphoproliferation, malignancy and allergy. These results are consistent with a recent report on initial manifestations in patients with PIDs where the majority presented with infections (77%) as well as immune dysregulation (18%) and malignancy (0.8%) [9]. The authors also described that 12% of their patients with IEI presented with syndromic features [9]. In the group of syndromic combined immune deficiencies, 56% initially presented with infections and/or immune dysregulation; but importantly, 89% of the remaining patients had syndromic features as their first presentation [9]. Patients with PIDs are prone to develop infections and non-infectious complications such as autoinflammation, autoimmunity, lymphoproliferation, malignancy and allergy [8,10]. We propose updating this schematic to include non-immunological syndromic features (Figure 2). Patients with PIDs have overlapping features and they are not dichotomised. This concept is useful in a PID diagnostic workup of patients who present in any of these forms. 

Our cohort shows a significant amount of preventable morbidity (e.g., bronchiectasis), adding to the argument that these children would benefit from early recognition and referral to a paediatric immunologist. Prompt diagnosis of IEI and appropriate management is associated with reduced morbidity and mortality [11]. A multidisciplinary approach to monitoring for complications, preventative management and definitive treatment should be appropriately instituted early with an aim to improve outcomes. 

## 5. Conclusions

Patients with sPIDs often have multiorgan involvement and some are non-immunologically mediated. There should be a low threshold to clinically assess and investigate for disorders of immunity in any patients with syndromic features especially when they present with recurrent/severe/opportunistic infections, features of immune dysregulation, or malignancy. 

## Figures and Tables

**Figure 1 jcm-12-04964-f001:**
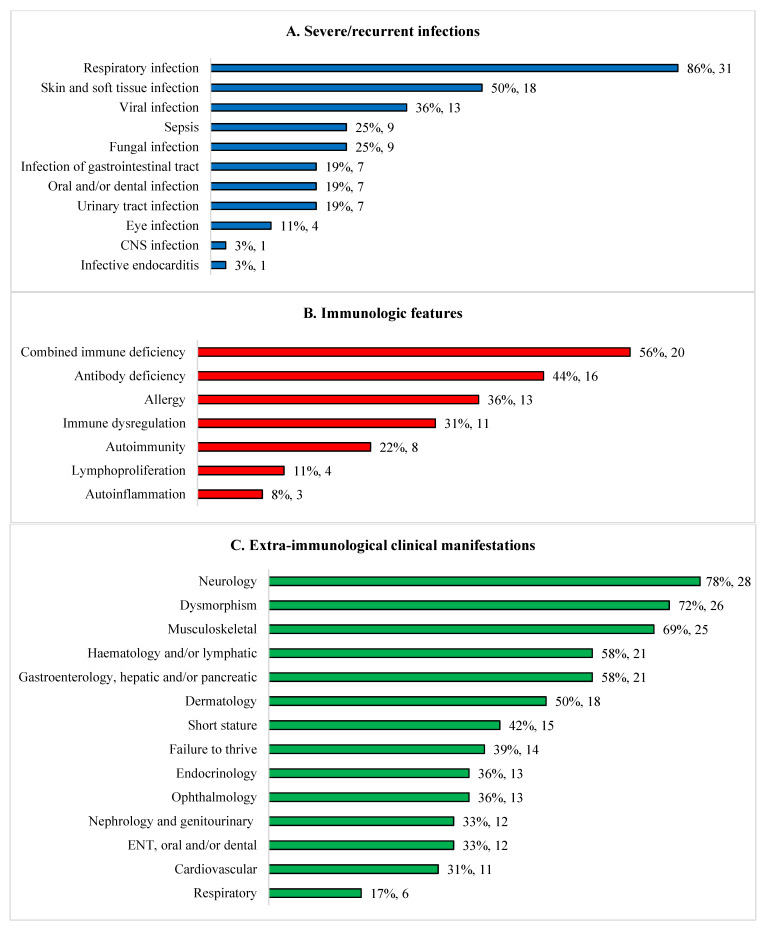
Clinical and immunological features of 36 children with syndromic immunodeficiencies. (**A**) Severe/recurrent infections. (**B**) Immunological features. (**C**) Extra-immunological clinical manifestations. CNS, central nervous system; ENT, ear, nose and throat.

**Figure 2 jcm-12-04964-f002:**
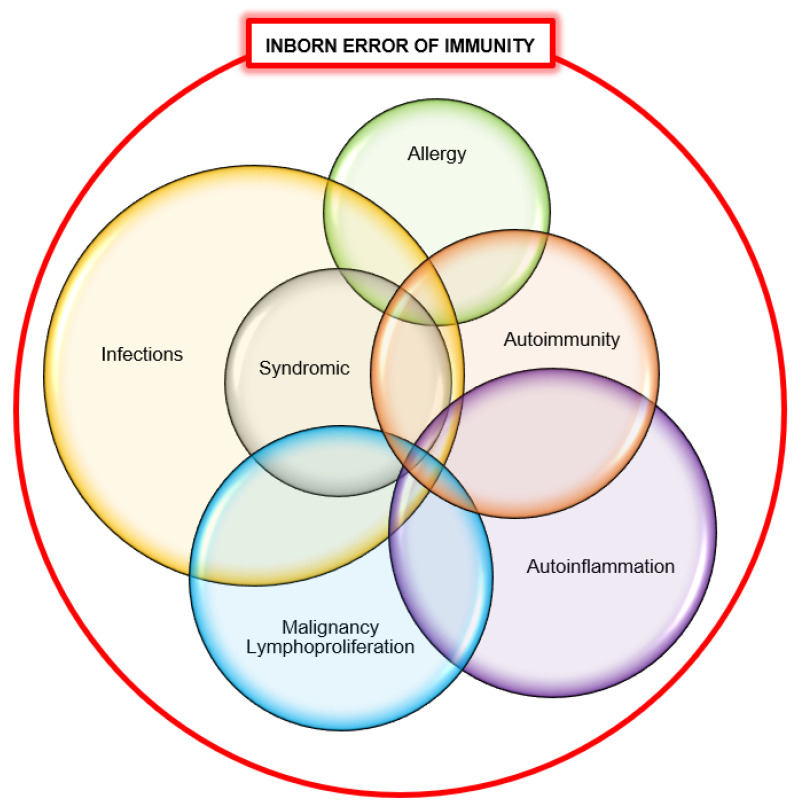
The diverse disease spectrum of inborn error of immunity (adapted from Walter et al.) [8].

**Table 1 jcm-12-04964-t001:** Immunologic features and treatment of 36 children with syndromic immunodeficiencies.

Syndromic Immunodeficiencies (n), Genetic Defect (OMIM Number)	Immunological and Clinical Manifestations (n) [1,2]										Treatment (n)		
	Combined Immune Deficiency	Antibody Deficiency	Innate Immune Deficiency	Recurrent/Severe Infections	Autoimmunity	Auto-inflammation	Immune Dysregulation	Lymphoproliferation	Malignancy	Allergy	Ig replacement	Antibiotic Prophylaxis	Immunosuppression
AT (n = 4), ATM (#208900)	4	1		2				2	2	1	4	2	
CHH (n = 1), RMRP (#250250)	1			1		1	1				1	1	
2p16.3 deletion (n = 1) (#614332) ^a^		1		1								1	
DiGeorge (n = 5), 22q11.2 (#188400)	5			5	5		2	2			5	5	4
Supernumerary ring chromosome 20 (n = 1) ^a,b^		1		1						1	1	1	
STAT3 deficiency (n = 4), STAT3 (#147060)	4			4						3	3	3	
Kabuki (n = 3), KMT2D (#147920)		3		3							1	2	
Arboleda-Tham (n = 1), KAT6A (#616268) ^a^	1			1							1		
Roifman (n = 2), RNU4ATAC (#616651)		2		2	1		1			1	2		
MIRAGE (n = 1), SAMD9 (#617053)		1		1								1	
Myhre (n = 1), SMAD4 (#139210) ^a^		1		1							1	1	
Noonan (n = 1), PTPN11 (#163950) ^a^		1		1						1	1	1	
SIFD (n = 3), TRNT1 (#616084)		3		3		2	2			1	3	1	3
THES (n = 2), SKIV2L (#614602)/TTC37 (#222470)	1	1		2			2			2	2	1	2
Trisomy 22 (n = 1) ^a,b^	1			1							1	1	
TTD/CS (n = 1), ERCC2 (#601675) ^a^		1		1							1	1	
WAS (n = 4), WAS (#30100)	3			3	2		3			3	3	3	1

AT, ataxia telangiectasia; CHH, cartilage hair hypoplasia; Ig, immunoglobulin; MIRAGE (myelodysplasia, infection, restriction of growth, adrenal hypoplasia, genital phenotypes, enteropathy); OMIM, Online Mendelian Inheritance in Man; SIFD, sideroblastic anaemia with B cell immunodeficiency, periodic fevers and developmental delay; STAT3, signal transducer and activator of transcription 3; THES, trichohepatoenteric syndrome; TTD/CS, trichothiodystrophy/Cockayne syndrome complex; WAS, Wiskott–Aldrich syndrome. Grey boxes indicate no patients with matching immunological, clinical manifestations and/or treatment. ^a^ Genetic condition not included in the 2022 International Union of Immunological Societies (IUIS) classification [2]. ^b^ OMIM number not available.

## Data Availability

Data are contained within the article or Appendix A.

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
