# Peer review of "The Multifaceted Syndromic Primary Immunodeficiencies in Children"

_jcm, 2023, doi:10.3390/jcm12154964_

Round 1

Reviewer 1 Report

Interesting paper about a poorly debated topic i.e  the complexity of syndromic features and immunological defects in children with syndromic primary immunodeficiencies With this regard the Authors report a descriptive study of a cohort of children with syndromic primary immunodeficiencies observed  from 2006 to 2021. The paper details the clinical manifestations of the cohort reporting  many comments useful for the early recognition of these patients. Similarly the conclusion about the need to investigate for disorder of immunity patients with syndromic features, presenting warning signs of immunodeficiency, immunodysregualton or lymphoproliferation is of interest for the scientific community. The manuscript is well written as well figures and tables.  I do not have complaints.

Author Response

Thank you very much for your complimentary comments. 

Reviewer 2 Report

1) In the section of Material and methods add a table with inclusion criteria

2) Are the 36 enrolled patients classified as IEI or sPID patients?

3) On line 72 in the results section, you write that 33 patients had a genetic diagnosis, but in the Table 1 a genetic diagnosis was reported for all 36 patients, so how have you classified the 3 patients without a genetic output?

4) Patients number with Wiskott-Aldrich syndrome and Ataxia telangiectasia are discordant between Table 1 and table S1

5) Do the grey squares mean that there are no patients matching that requirement?

6) The figure 2 does not represent all intersections between the different conditions, for example patient with Trisomy 22 has a syndromic condition together with recurrent and severe infections, but in the figure 2 the two corresponding circles do not intersect each other

7) Why have you highlighted those 10 patients in the Table S2? Moreover there are 12 patients reported in Table 1 with immunosuppressive therapy and only 10 in Table S2

Author Response

Point 1: In the section of Material and methods add a table with inclusion criteria 

Response 1: Thank you for this suggestion. We had added inclusion criteria in bullet form instead of creating another table (line 67-78 of page 2).

Point 2: Are the 36 enrolled patients classified as IEI or sPID patients?

Response 2: The patients enrolled were classified as sPID. To clarify this, we made minor changes to the sentence in line 82 of page 2.

Point 3: On line 72 in the results section, you write that 33 patients had a genetic diagnosis, but in the Table 1 a genetic diagnosis was reported for all 36 patients, so how have you classified the 3 patients without a genetic output?

Response 3: There was no information from medical record to indicate/estimate the age of genetic diagnosis in 3 patients, therefore we could only report the age of 33 patients when they were first diagnosed with genetic disorders.

Point 4: Patients number with Wiskott-Aldrich syndrome and Ataxia telangiectasia are discordant between Table 1 and table S1

Response 4: Thank you for identifying this mistake. Table S2 is correct. We have changed the numbers of patients with ataxia telangiectasia and Wiskott-Aldrich syndrome in Table 1.

Point 5: Do the grey squares mean that there are no patients matching that requirement?

Response 5: This is correct. An explanation has been added into the footnote of the table.

Point 6: The figure 2 does not represent all intersections between the different conditions, for example patient with Trisomy 22 has a syndromic condition together with recurrent and severe infections, but in the figure 2 the two corresponding circles do not intersect each other.

Response 6: Figure 2 has been changed to reflect the reviewer’s comment and to closely resemble original figure from Walter et al [8] but with additional “syndromic” bubble.

Point 7: Why have you highlighted those 10 patients in the Table S2? Moreover there are 12 patients reported in Table 1 with immunosuppressive therapy and only 10 in Table S2.

Response 7: The 2 patients with ataxia telangiectasia had chemotherapy related immunosuppression for the treatment of lymphoma and they were excluded from table S2. We have decided to remove them from Table 1 to avoid confusion.
